# The VIRTUALDiver Project. Making Greece's Underwater Cultural Heritage Accessible to the Public

**George Pehlivanides** [1], **Kostas Monastiridis** [1], **Alexandros Tourtas** [1], **Elli Karyati** [1], **Giotis Ioannidis** [1], **Konstantina Bejelou** [2] , **Varvara Antoniou** [2] and **Paraskevi Nomikou** [2,*]

[1] Tetragon S.A., 54641 Thessaloniki, Greece; interaction@tetragon.gr (G.P.); develop@tetragon.gr (K.M.); info@tetragon.gr (A.T.); expo@tetragon.gr (E.K.); giotis@tetragon.gr (G.I.)

[2] Department of Geology and Geoenvironment, National and Kapodistrian University of Athens, 15784 Athens, Greece; bejelouk@geol.uoa.gr (K.B.); vantoniou@geol.uoa.gr (V.A.)

* Correspondence: evinom@geol.uoa.gr; Tel.: +30-2107274865

**Abstract:** Reaching the underwater world is undoubtedly an incomparable adventure. Impressive geological structures, flourishing ecosystems, shipwrecks, and submerged landscapes lie beneath the sea surface in wait for discovery. However, this world is accessible only to those who have the chance to dive or to scientists conducting underwater research. By means of a dynamically developing sector of informatics utilizing Virtual (VR) and Augmented Reality (AR) practices, the VIRTUALDiver project intends to provide access to all the aforementioned hidden "treasures" through the creation of an innovative platform providing unique interactive experiences. More specifically, specialized guided tours in natural and virtual environments covering areas of touristic, cultural, and environmental interest. VIRTUALDiver is an experience, design and content presentation platform, a custom-made add-on environment within the Unity 3D authoring tool, offering the ability to manage multimedia content in a simplified way. No specialized programming knowledge is required, enabling the project's interdisciplinary consortium to easily collaborate and exchange ideas. The expected result is the establishment of a successful educational and entertaining cultural product to support businesses and professionals operating in the field of culture-tourism. Above all, VIRTUALDiver aspires to become a novel form of storytelling, immersing the user into unique experiences under the waves.

**Keywords:** virtual reality; augmented reality; underwater cultural heritage; Unity 3D

## 1. Introduction

The underwater environment in Greece is of great scientific interest in all fields of marine research. However, it has not yet been sufficiently exploited for cultural and tourism purposes. At the same time, technological advantages in the fields of Virtual (VR) and Augmented Reality (AR) have undergone considerable development by providing technical solutions for environments that their modeling has been problematic or non-operational in the past. From medical training to gaming, these technologies provide a variety of prospects that can lead previously designed and applied schemes to new directions and help new ideas to arise and develop. Especially in the area of cultural management, creating digital narratives accessible not only to the experts but also and, most importantly, to the general public in an intriguing and scientifically accurate way is of great significance [1–6]. What makes the narration in a virtual or augmented environment special is the feeling that the user is "living" the experience, allowing the narrative to be perceived through different perspectives or characters (avatars), while the "immersion" stimulates more emotional involvement. That is why for the past decade a number of relevant projects have been running towards successful cultural implementations [7].

The Underwater Cultural Heritage (UCH)—a substantial cultural capital that lies on the seabed of the world—is an important resource that needs to be protected and valued. It also stands as a triggering factor for touristic development, concerning especially underwater archaeological sites, be they shipwrecks or submerged landscapes, highly fascinating for the public, both for the sense of mystery that surrounds them and the symbiosis between artifacts and sea life. Access to these sites is limited, as only a small percentage of people can actually dive or due to the fact that a great number of these sites are in depths beyond human reach. Digital technologies (e.g., Virtual Museums, Virtual Guides, and Virtual Reconstruction of Cultural Heritage, Augmented Reality applications) provide a unique opportunity for digital accessibility to both scholars and the general public, interested in having a broader perception of these underwater sites [8–11]. Some representative cases of applying VR and AR technologies in the promotion of UCH are shown here in order to place the framework of the VIRTUALDiver project. Project VISAS: Virtual and Augmented Exploitation of Submerged Archaeological Sites, is a collaborative research project created to improve the responsible and sustainable exploitation of underwater archaeological sites by reconstructing the underwater terrain, creating a VR system for exploring this environment, and making a virtual guide experience for mobile devices used underwater [12,13]. Researchers of the project i-MARECULTURE integrated digital technologies to promote UCH to the public [14]. They developed a VR system that allows users to experience underwater environments with high cultural value, using gamification methods to create two serious games that explained the underwater excavation methods. Additionally, in order to improve underwater cultural tourism, they developed a waterproof AR system capable of acting as a guide of specific underwater archaeological sites. Other projects with similar research focus are the Lab4Dive [15,16] and the VENUS project [17,18], both of which are scientific platforms that facilitate underwater research and, ultimately, communicate UCH to other scientists and the public. Apart from the above academic research, projects from the private sector are also worth mentioning. The Blu: Deep Rescue is a virtual underwater experience giving the participants the opportunity to swim together with sea creatures like turtles, sharks and whales [19]. The Ocean Maps company has created a series of applications for virtual diving, specially designed for divers, with scheduled routes providing additional information about the underwater terrain and sea life [20]. The Underwater Virtual Museum—Underwater Malta, is a platform that gives the opportunity to investigate shipwrecks around Malta in a Virtual Reality environment [21]. Finally, the company 360° Virtual Tour Co. has created the Underwater Coral Reefs in 360°, where the user can navigate through coral reefs in Palmyra Coral Gardens using 360° photography technics [22]. The bibliography on such matters is growing rapidly [23–30].

In almost all of the above cases, VR "dry dive" experiences were designed using all available technological and methodological tools in order to provide an immersive cultural experience that would engage the user with as many of the stimuli of a diving visit as possible.

In this context, the VIRTUALDiver project was developed as an integrated interactive platform for exploring natural (AR) and virtual (VR) environments in areas of touristic and environmental interest (e.g., unique geomorphological structures, shipwrecks, sunken harbors and parks, marine parks, NATURA areas, etc.) in order to enrich travel experience and promote specific and diverse forms of tourism. The language compatibility of the platform in its pilot version will be English. Partners from various backgrounds were engaged in this endeavor. The Faculty of Geology and Geoenvironment [31], leading this research project, is a part of the School of Sciences of the National and Kapodistrian University of Athens. The aim of all undergraduate and postgraduate programs of the Faculty is to develop an enthusiasm for the geosciences and environment, thus as to educate students on meeting scientific challenges through a learning experience provided by academic personnel with an important role in the international research forefront and a wide variety of research activities with emphasis on excellence, innovation, and interdisciplinarity. TETRAGON S.A. [32] specializes in a broad spectrum of fields, such as innovative product research and development, digital and interactive applications, audio-visual productions, methodologies, services and technologies for the documentation, showcasing,

and promotion of cultural heritage and tourism, as well as consulting, design, and construction of cultural spaces, museums, and environmental infrastructure. Up2metric P.C. [33] develops innovative custom software solutions and provides consulting engineering services in the fields of computer vision, photogrammetry, machine learning, remote sensing, and metrology. Steficon S.A. [34] is a 360° global full-service digital agency that manages Internet brands, consisting of a team of long-lasting professionals with fusion skills such as creative thinking and design, digital and performance marketing, development of high-end ICT web, mobile applications, and video production.

All the above collaborated in order to provide a successful educational and entertaining cultural product. Significant, multi-temporal, validated knowledge, information and data from the university partner guarantee a detailed, high-resolution mapping of the underwater and terrestrial relief, which was carried out with state-of-the-art technologies (swath mapping systems, underwater vehicles, unmanned aerial vehicles), creating a synthetic topographic relief basemap and analyzing all its particular geomorphological and ecological structures as well as all anthropogenic interventions. Through a set of specially designed tools for multimedia content management with an emphasis on green screen photography and 360° video production, the design team is able to write narrative scenarios and produce interactive experiences for VR and AR environments. VIRTUALDiver is targeting to address the sector of virtual tourism and culture and invest in this opportunity with regard to the current industrial competition in Greece and Europe. An additional goal is the development of new knowledge and skills, more specifically the integration of research knowledge into interactive narrative systems that will lead to the establishment of innovative tourism products, in order to promote the complex, challenging, magnificent underwater environment and to attract tourists of general or special interest. The research results will be communicated through the developed platform, while the systematic data collection methodology and the virtual representation will be scalable and applicable to any other coastal environment of touristic and/or cultural interest.

## 2. Interaction Components and Methods

The VIRTUALDiver was a custom-made, add-on environment within the Unity 3D authoring tool [35], providing a vast set of interaction design methods for content presentation and storytelling.

Virtual Diver Platform handles high-quality and resolution assets like 3D objects, videos, and images. Besides that, the goal was to provide seamless interaction with end-users. These requirements helped us set the hardware specifications both for the interaction with the Virtual Diver Platform as well as the exported experience. As a minimum set-up according to the current technological standards, the following specifications were suggested: Processor Intel Core i7-8700 [36] or AMD Ryzen 7,2700 [37], Graphics Processing Unit: NVIDIA GeForce RTX 2080 [38] or NVIDIA Quadro RTX 6000 [38], Main Memory: 32 GB, Storage: NVMe SSD 250 GB, Head Mounted Display: Oculus Rift S [39] or better, Operating System: Windows 10 (64-bit) [40].

Utilizing the scripting capability of Unity 3D, a series of algorithms (scripts) were developed as the programming basis for creating special tools for interaction with multimedia content (Interaction Components). The interaction tools were grouped in a library that allows the design team to implement multimodal interactive experiences through narrative scenarios. The implementation of the Interaction Components was divided into 2 sections. The 1st section concerned the creation of tools (editor-scripts) specially designed for use within the Unity 3D environment in order to assist the design team in creating and enriching various Points of interest (POIs) with multimedia content, thus creating a database (scriptable object) within the platform. The 2nd section concerned the development of User Interface (UI) representational tools that will communicate with the database and present the multimedia content to the end-users of the narrative experiences in the best possible way.

The 1st step in creating the Interaction Components was to determine the requirements and specifications of each Component, as well as what kind of multimedia it would deal with. Specialized media types were considered, such as the use of high-resolution 3D models [41], high-resolution 360° spherical video playback, 360° virtual tour reproduction using high-resolution

photographic equipment [42], the ability to reproduce spatial 3D audio, etc. Following that, the needs of the basic functions of the interaction tools were identified. Taking into account the above requirements and utilizing the capabilities of Unity 3D authoring tool, a Scriptable Object was created, which acted as the database of the narrative's POIs. Scriptable Objects are databases that can contain large amounts of data by reducing memory usage of the project, as data copying is limited. The design team adds the Interaction Components corresponding to the multimedia they want to present to the Game Object that represents the POI and forms the way it will be presented as well.

Regarding the creation of the Interaction Components' interfaces, in the Unity Editor, the emphasis was placed on simplified design and ease of use by people with varying degrees of familiarity with Unity 3D authoring tool.

The following functional requirements were considered throughout the design process of the interface of the Interaction Components in the Unity editor (Figure 1):

(1)　Proper visualization of the type of media that the tool will manage through a representative icon (e.g., image thumbnail icon),
(2)　The use of a title and a related auxiliary summary that will explain the properties of the POI, and
(3)　The selection and attachment of the interaction tools through drop-down menus in order to form the narrative scenarios for each landmark.

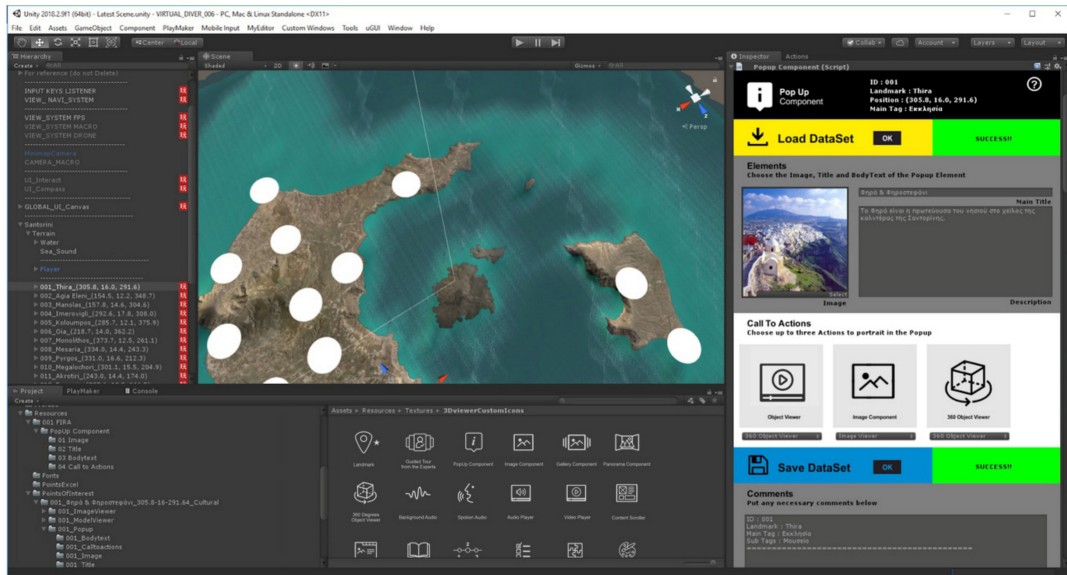

**Figure 1.** Screenshot from Unity Editor—Image Viewer Interaction Component Editor (on the right side of the screen).

Interaction components' interface, in the Unity Editor, comprises 5 distinct sections, which correspond to relevant steps of the workflow (Figure 2):

- Section 1: Basic information about the tool and the POI, such as tool type, unique POI code, name, coordinates, and basic tag, as well as the instructions for use.
- Section 2: Enables the automatic import of data from the data collection and processing stage.
- Section 3: This is the main part of the tool and provides the ability to insert new data as well as to configure the existing one. Concerning the Image Component tool, photos are inserted, and provided fields correspond to the photo, title, and description.
- Section 4: Enables the function of saving changes to the database.
- Section 5: Consists of a comment field. Comments are stored in the database and are visible to all members of the design team who use the platform.

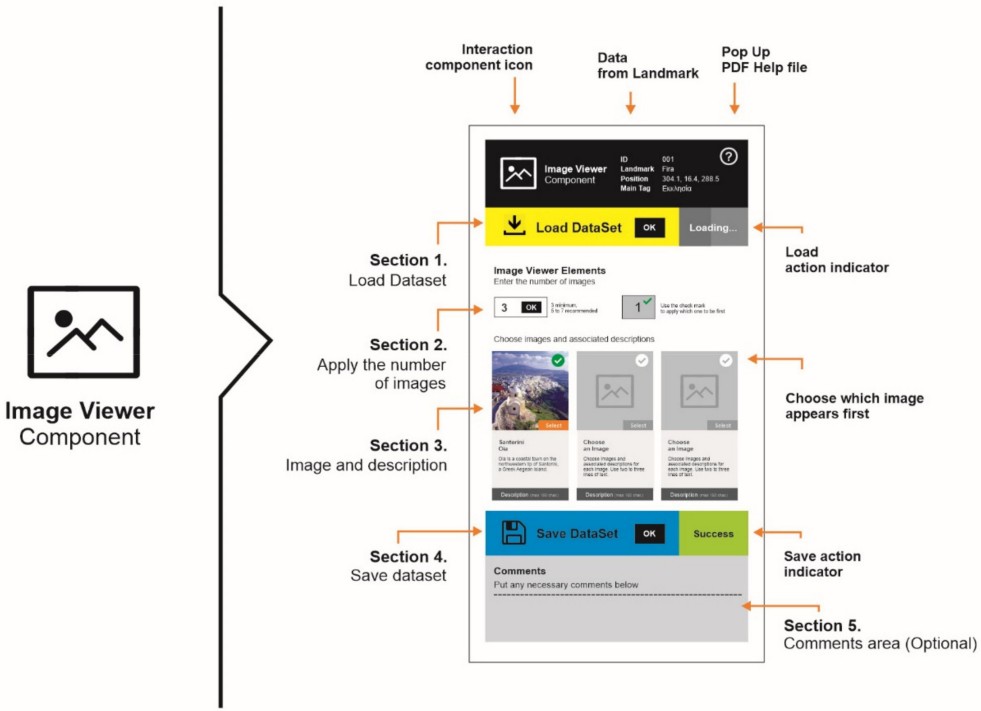

**Figure 2.** Image Viewer Interaction Component User Interface—showing the distinct sections of the interface.

The way the available content is displayed depends on the specific presentation template (design template) as well as the medium (VR goggles, desktop environment, tablet) that will be used (Figures 3–5). The following design principles are followed:

- Consistency in the presentation of content in various media.
- Design solutions that facilitate the ways of interaction in both VR glasses and a conventional computer screen, as well as in smaller screens.
- Color combinations that facilitate the readability of graphics for use in various media.
- Selection of typographic elements, font family, and auxiliary icons that offer optimal readability and ease of interaction in various media.

Regarding navigation and interaction in the virtual environment, the Virtual Reality Tool Kit (VRTK) [43] is used, a complete open-source package for the Unity 3D authoring tool that offers high-quality ready-made solutions for navigation systems, interacting and manipulating virtual objects for interactive experiences in VR. For the development of the VIRTUALDiver platform, the VR glasses system, Oculus Rift S [44], has been used.

Due to the interdisciplinarity of the design team and the goal to create a platform that will be able to be used by experts of different backgrounds, 2 specific Unity plugins were used to ease the design process. The first one was a visual scripting tool, Playmaker [45], especially designed for Unity 3D. Through visual scripting, the team was able to manipulate and graphically connect the various functions of the platform [46]. The Playmaker add-on software was specially used to visually program the functions of each UI element through pre-existing or custom-made actions (Figure 6). The 2nd tool, selected amongst others as the best for the specific task, was the PSD2GUI plugin [47], and was used in order to instantly import Graphical User Interface (GUI) design templates from Photoshop [48] files into Unity 3D. This tool recognizes the layers into a Photoshop file and turns them into usable interaction objects inside Unity 3D. With the use of PSD2GUI the user interface was instantly transferred into Unity 3D, a fact that saves time and effort for the design team.

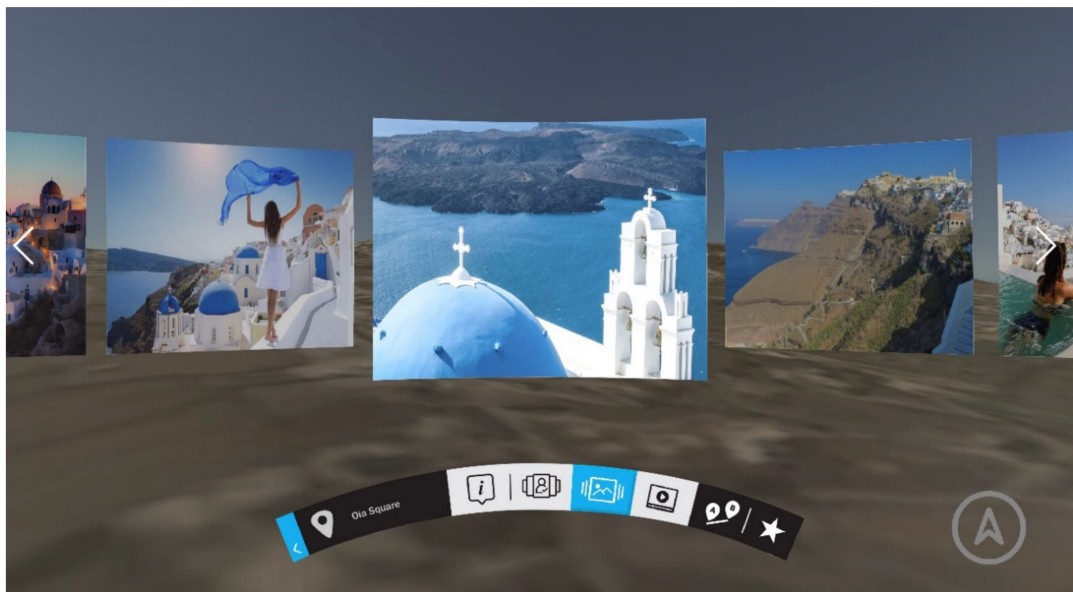

**Figure 3.** Multimedia selection in VR environment—Image Viewer Component showing different views (photos) of Santorini island.

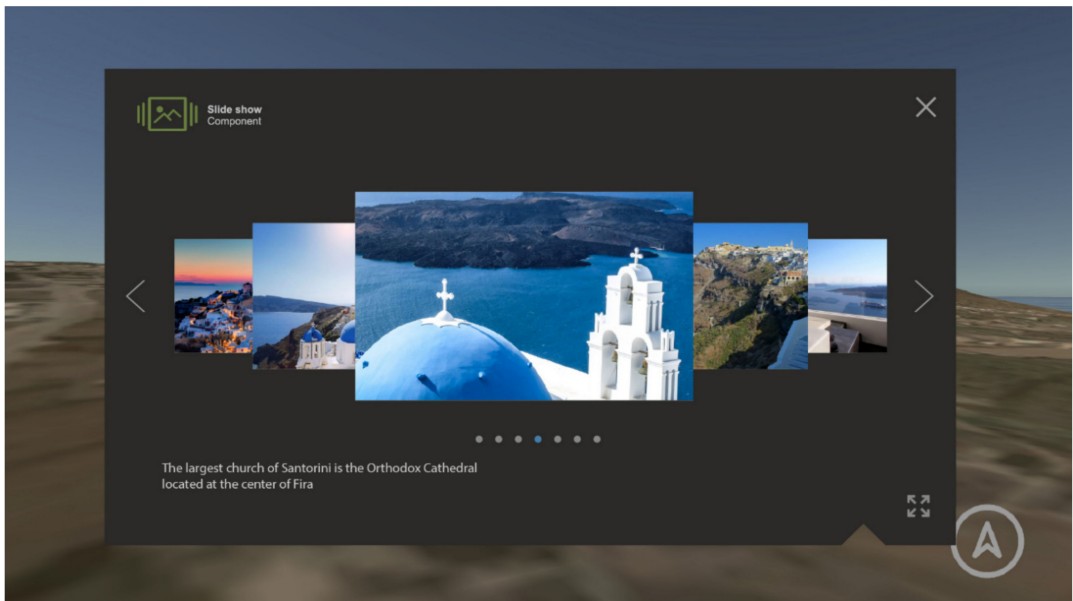

**Figure 4.** Multimedia selection in Desktop VR mode—Image Viewer Component showing different views (photos) of Santorini island.

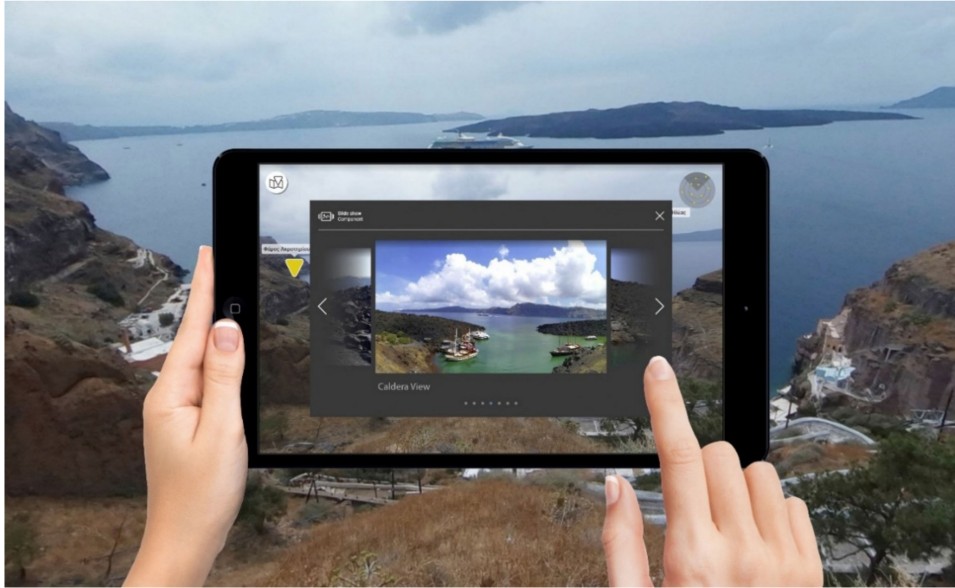

**Figure 5.** Multimedia selection in augmented reality (AR) environment—Image Viewer Component showing different views (photos) of Santorini island.

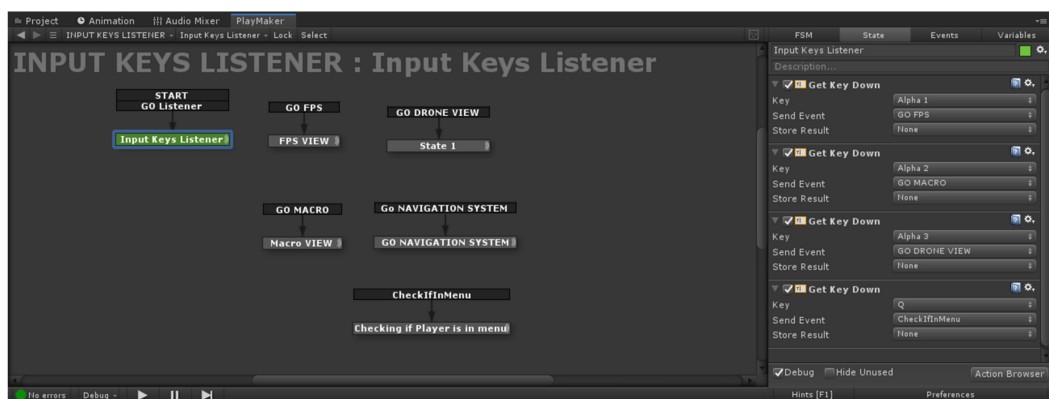

**Figure 6.** Playmaker—Visual Scripting Tool (User Interface Screenshot).

Specifically, for the Virtual Environment, a special tool was created to help the end user navigate between the different Interaction Components. This tool consisted of an ergonomic interaction bar (Figure 7) that was placed in the optimal position, in front of the user, giving him/her flexibility of movements. Interacting with the functional elements of the bar, the user retrieves information from various multimedia content provided by the POI. The different states of each button (active/idle) were visually stated, thus that the user was constantly informed about the component he/she was dealing with. This tool has been created to provide the easiest and most comfortable experience to the user, based on previous studies like the Oculus Virtual Desktop [49]. The bar was designed in Photoshop following the proposed Unity 3D plugin methodology, PSD2GUI, through which it was integrated into Unity 3D in a World UI Canvas format, ready for editing. Subsequently, an automated system was designed that communicates with the database and has the ability to display the respective Interaction Components of each POI.

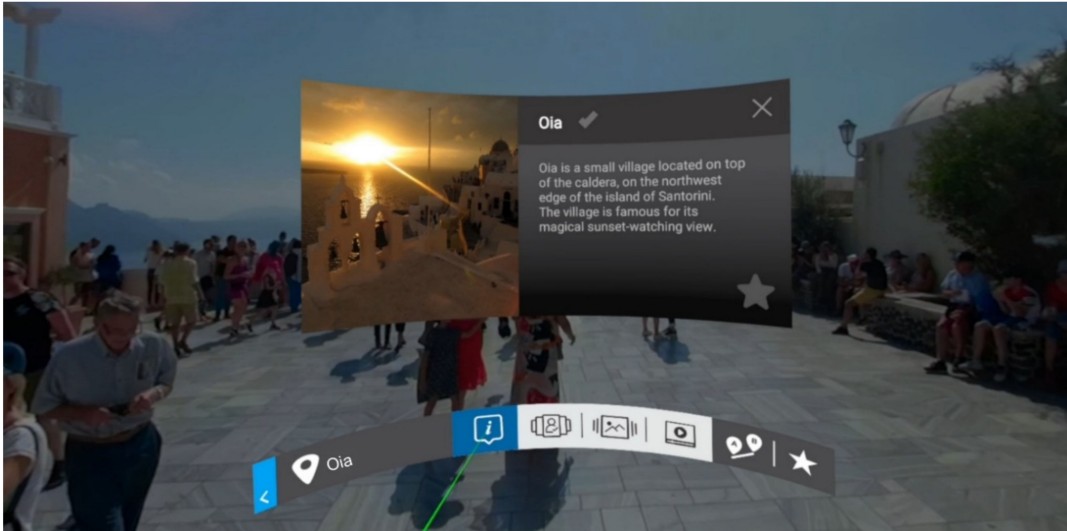

**Figure 7.** Navigation bar in VR environment—Oia Square. Active Interaction Component: PopUp Component indicated on the navigation bar with blue color.

## 3. Results and Discussion

The VIRTUALDiver platform is a complete solution for the design, management, and implementation of interactive experiences for guided tours in a natural and virtual environment in areas of touristic, cultural, and environmental interest. The platform utilizes the capabilities provided by Virtual (VR) and Augmented Reality (AR) technologies to create an innovative tool in order to support businesses and professionals operating in the field of Culture-Tourism.

### 3.1. The Platform

#### 3.1.1. Interdisciplinarity

The VIRTUALDiver project aims at strengthening special forms of tourism, such as: Cruise tourism, diving, science tourism, and others. The platform was based on the interdisciplinary collaborative relationship of experts with the ultimate goal of creating interactive experiences through a scientifically sound approach (interdisciplinary and transdisciplinary design process). The main researchers (with theoretical and technological background) who make up the design team of VIRTUALDiver belong to the following scientific fields:

1. Scientific disciplines of geology,
2. Humanities with an emphasis on culture (archaeology, history, museology, etc.),
3. Disciplines of theoretical and practical issues of design, audio, and video arts,
4. Information and communication technologies,
5. Other scientific fields that may make a useful contribution, depending on the design strategy.

The interdisciplinarity of the consortium ensures the holistic character of the approach making the VIRTUALDiver platform a pioneering product in the market.

#### 3.1.2. The Product

The platform will be used by the consortium's experts as a specialized service in both technical and creative means, aiming at the formation of a new tourist product, unique in the international level, as it is based on the huge cultural reserve of the Greek sea, increasing the value of each institution-user (municipalities, ministries, cultural institutions, etc.) and the assets (natural-cultural) of the area to which it will be provided. As a complete design solution for the management and implementation

of interactive experiences, it consists of distinct subsystems, which are accessible depending on the capacity and rights of use by: (a) The respective cultural project assignee (e.g., municipality, cultural organization) that will provide the necessary data and specifications, as well as (b) the design team that undertakes the design and production of the interactive experiences for the selected medium.

### 3.1.3. System's Architecture

The system's architecture (Figure 8) comprises two subsystems:

(1)　The data collection subsystem is based on internet technologies and aims at creating, processing, and maintaining the data prototypes required for subsequent use by the interaction and experience design subsystem. Through this subsystem, the user is able to upload all the necessary multimedia material to a database in order to create interactive experiences.

(2)　The subsystem of interaction, experience design, and content presentation uses the capabilities provided by Unity 3D development platform, to design and present interactive experiences. For this purpose, techniques and tools for interaction with multimedia content are developed to facilitate the design team in the production process of interactive narratives. Through simplified procedures, the design team is able to incorporate the necessary multimedia initially imported in the data collection subsystem, as well as the respective geomorphological terrain into the software. As a result, the design team is able to create various types of interactive experiences based on pre-designed paths. Depending on the design that is followed for the interactive experience, the final result can be exported to the corresponding medium (VR goggles, tablets, etc.). The key element of the platform is that it provides mechanisms for the implementation of interactive experiences with dynamic content, separating the collection and configuration of multimedia material from the development and visualization of the interactive experience. It also enables the sustained updating and processing of the multimedia material by the design team, according to the requirements and needs of the customer-user as well as the design specifications that have been set or updated. The subsystem of interaction, experience planning, and content presentation provides the design team with the following capabilities:

- Multiple ways of interacting with the content, such as slideshow, panoramic photos, videos, audio clips, text, timeline, specially designed museo-pedagogical applications, etc.
- Multiple ways of presenting the content, such as Virtual Maps, Routes, Geographical Areas, Through Thematic Categories, etc.
- Future expansion and additions, without necessary changes in the existing material.

The designed tools, mentioned above, are the Interaction Components, and have the ability to utilize the multimedia material gathered in the database. A series of interaction components are added to each Point of Interest (POI), thus giving it a dynamic substance for each defined narration. These tools offer the ability to manage multimedia content in a simplified way without requiring specialized programming knowledge (Figure 9).

The Interaction Components offer the possibility of updating/adding new ways of interacting with supervisory content both in different media (virtual reality glasses, tablets, etc.), as well as in different environments (VR, AR, standalone desktop applications, etc.). As a whole, they form the Library of Interaction Components, where they can be combined and create interactive experiences in the narrative scenarios of the selected routes.

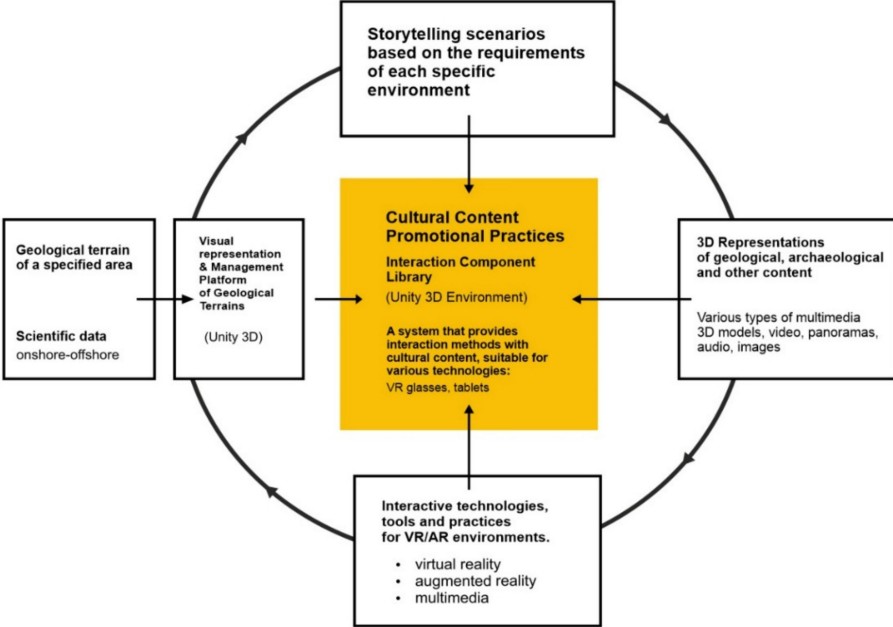

**Figure 8.** Overall system architecture.

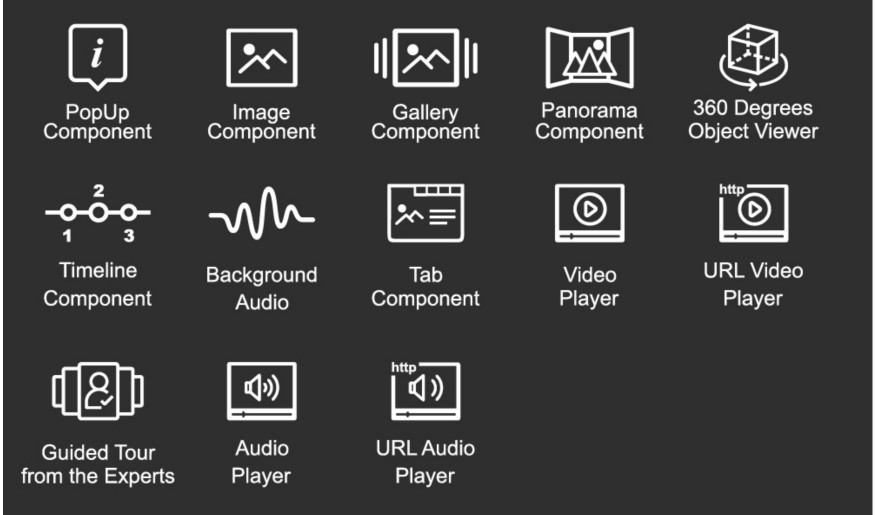

**Figure 9.** The Interaction Component Library: List of Interaction Components created for the VIRTUALDiver platform accompanied by their respective icons.

### 3.2. Santorini Case Study

The platform is piloted for the submarine area of the Santorini caldera, but also for the Mediterranean underwater volcano, Kolumbo (Figure 10), and enables users to navigate virtually in environments accessible only by underwater vehicles and in cost-intensive research and scientific missions. Santorini is strategically targeted as one of the most touristic destinations in the world. The uniqueness and importance of the island are due to the active volcano, which has a history of 2.5 million years. Throughout its geologic record, several volcanic events sculptured Santorini, with the last major eruption, the Minoan eruption, having formed the current caldera 3600 years ago. The unique geomorphological scenery of the inner slopes of the caldera, which continues on the seabed (Figure 11), the particular settlement development, as well as the geological history and evolution of the volcano are the main themes of the created interactive narrative.

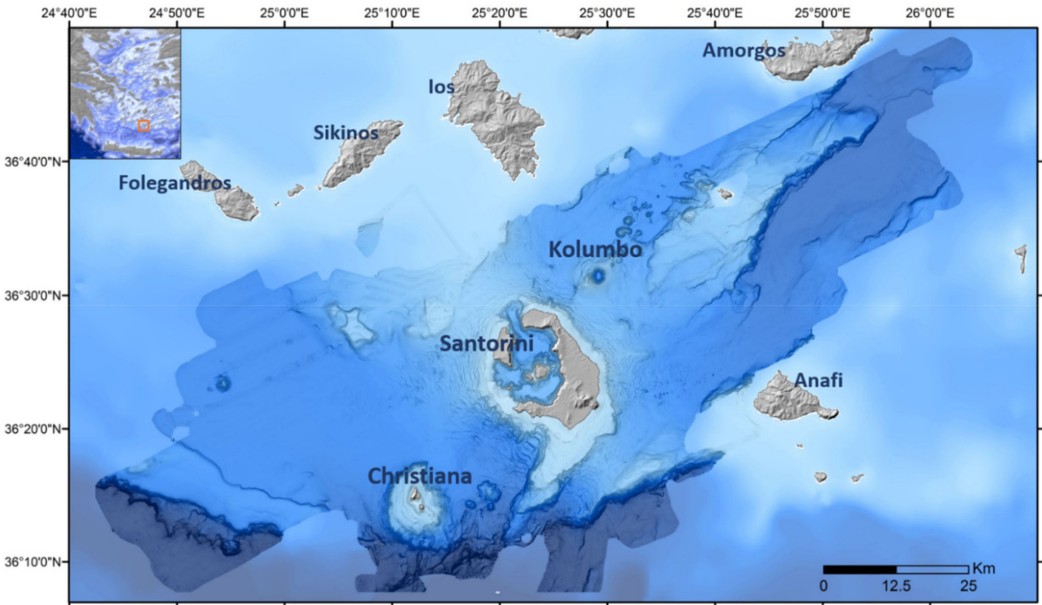

**Figure 10.** Map showing the exact location of the Santorini volcanic field within the Aegean Sea. Modified by Hooft et al., 2017 and Nomikou et al., 2013 [50,51].

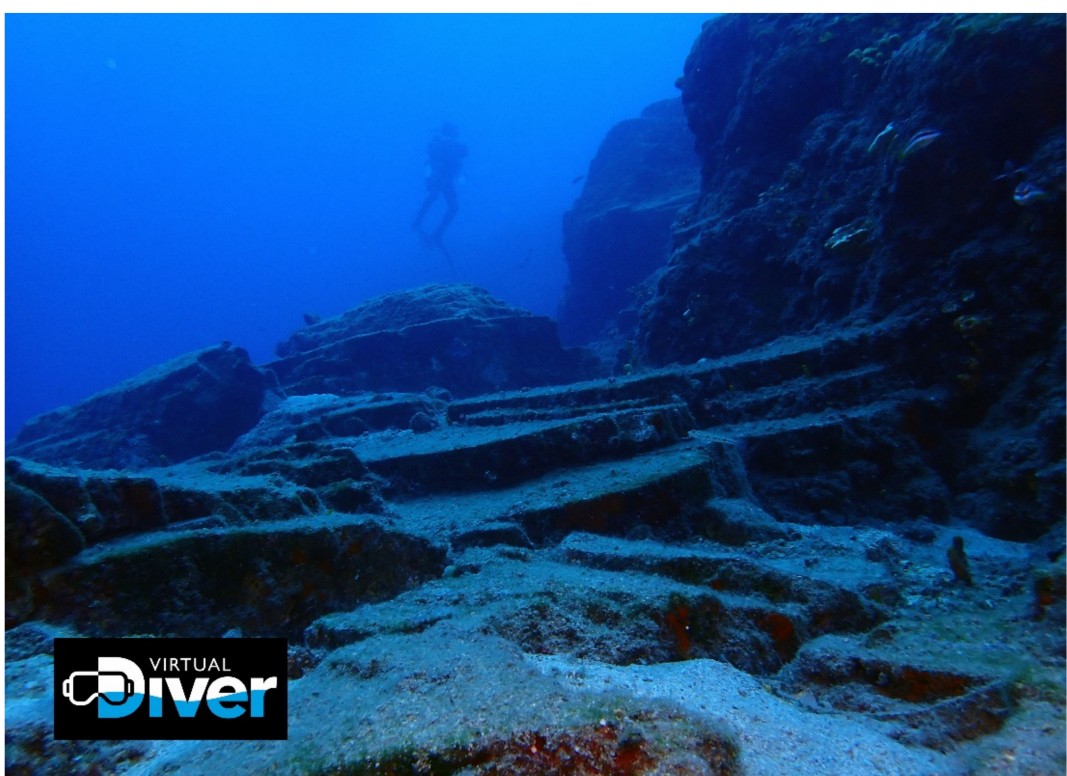

**Figure 11.** Underwater photo, case in point of the marvelous underwater volcanic character of Santorini region, showing the "Atlantis steps" at the submarine area near Thirasia island (credits to Othonas Vlasopoulos).

Specific user scenarios were created, highlighting the cultural heritage of the island using the Interaction Components that were developed for the platform. The scenarios include specific routes with stops where the user is able to receive multimedia material in various ways. The experience consists of three navigation modes: (a) Fly, (b) Walk, and (c) Diver Mode. In each mode, the user can navigate differently. Fly Mode acts as a way for the user to navigate over Santorini and aims to give

him/her the freedom to choose the POI he/she would like to interact with. For the development of the navigation system in Fly Mode, a system was developed that combines the sensors of the Oculus Rift S with the controls and tools provided by VRTK. The user targets the controllers in the direction he/she wants to move and activates the push of the virtual avatar towards it [52]. The speed for this movement is regulated in order not to provoke discomfort and dizziness to the user [53]. The user interacts with the POI using the VR controls combined with a pointer. Walk Mode allows the user to browse the Interaction Components and the multimedia content in the POI and move through the stages of each scenario. Finally, Diver Mode is a special case of navigation in an underwater environment, where the user can navigate and interact with. There, he/she can be informed about the flora and fauna of the seabed, examine shipwrecks, and geological phenomena, as well as obtain relevant multimedia content (see Supplementary Materials).

Initializing the application, the user has the option to navigate between different scenes, routes, and guided tours that are available by the platform (Figures 12 and 13). This option is added in order to ease the user's navigation through different POIs and is accessible at any time during the experience. In the example of the Oia-Kolumbo scenario, the user starts in Fly Mode (Figures 14 and 15), where he/she flies over Santorini. For the representation of the terrain, a special three-dimensional model is used, which is the result of research and standard data processing. To improve realism and, consequently, the user's immersion, water, volumetric clouds, high-resolution Skybox as well as the necessary soundscapes were added. Using the controls of the VR glasses, the user has the ability to fly over the island. Each POI is represented by a 3D Symbol accompanied by its interaction components for the user to identify and interact with. World UI Canvas, a native Unity 3D tool, is used in order to place UI anywhere inside the 3D space. As the user interacts with the POI, he/she is transferred in the Walk Mode environment.

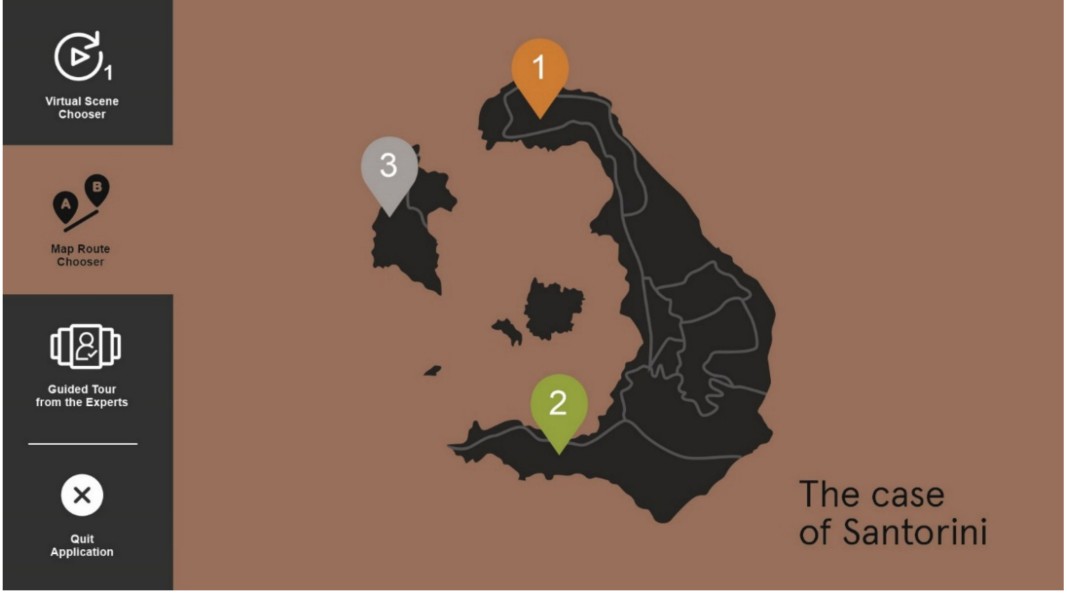

**Figure 12.** User Interface of Virtual Scene Chooser (screenshot): The user can choose between different scenes through a quick menu.

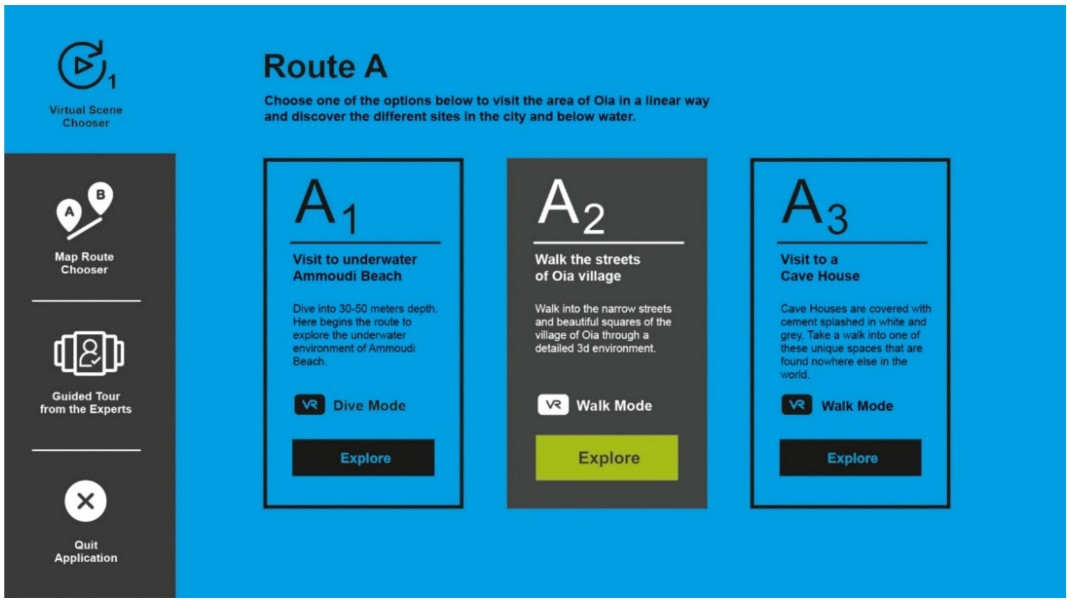

**Figure 13.** User interface of Map Route Chooser (screenshot): The user can choose between different routes through a map.

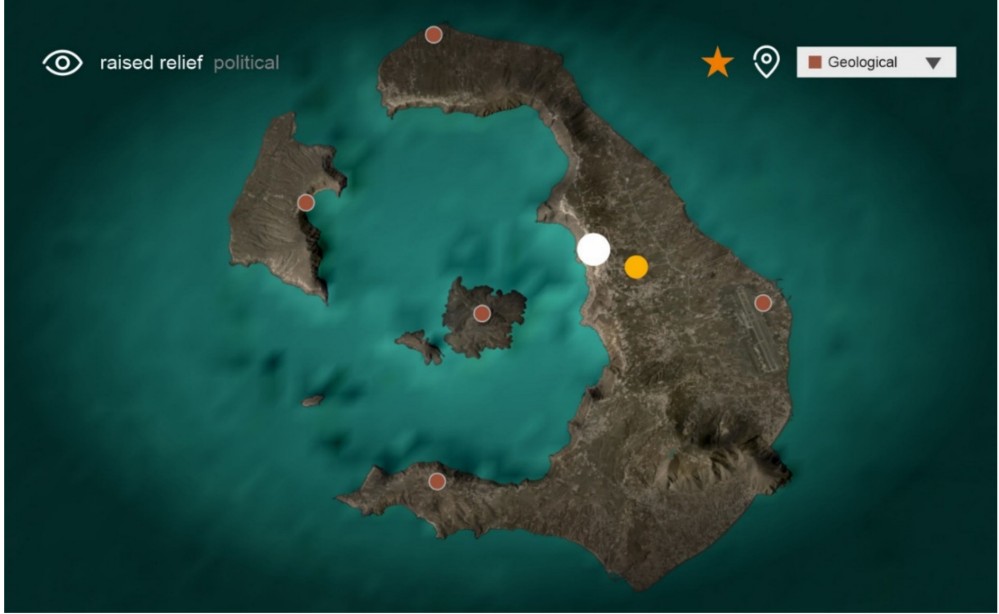

**Figure 14.** Top View of Santorini with marked Points of Interest (POIs) (White mark: City of Fira, Yellow mark: User's location, Red mark: Points of geological interest. 3D Model has been produced by Up2metric VIRTUALDriver partners.)

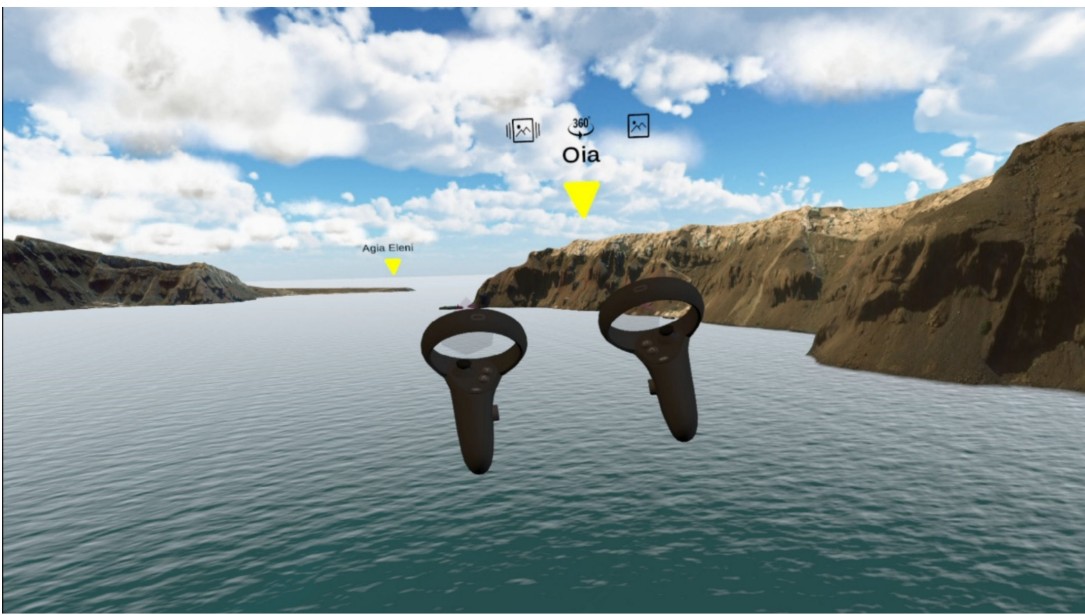

**Figure 15.** Fly Mode: User can navigate onto 3D terrain of Santorini, locate POIs and select them to enter Walk Mode. (3D Model has been produced by Up2metric VIRTUALDriver partners)

In the Walk Mode (Figures 16 and 17), depending on the structure of the scenario, the user is either in a 3D environment or in the center of a 360° video. In this particular scenario, the user is inside a 360° video of the central square of Oia. The user stays stable while the video plays on a loop. Using the controls, different multimedia content can be retrieved by choosing the different Interaction Components located on the interaction bar, allowing the user to view additional information about the POI. Selecting an Interaction Component displays the corresponding UI, with which the user can interact.

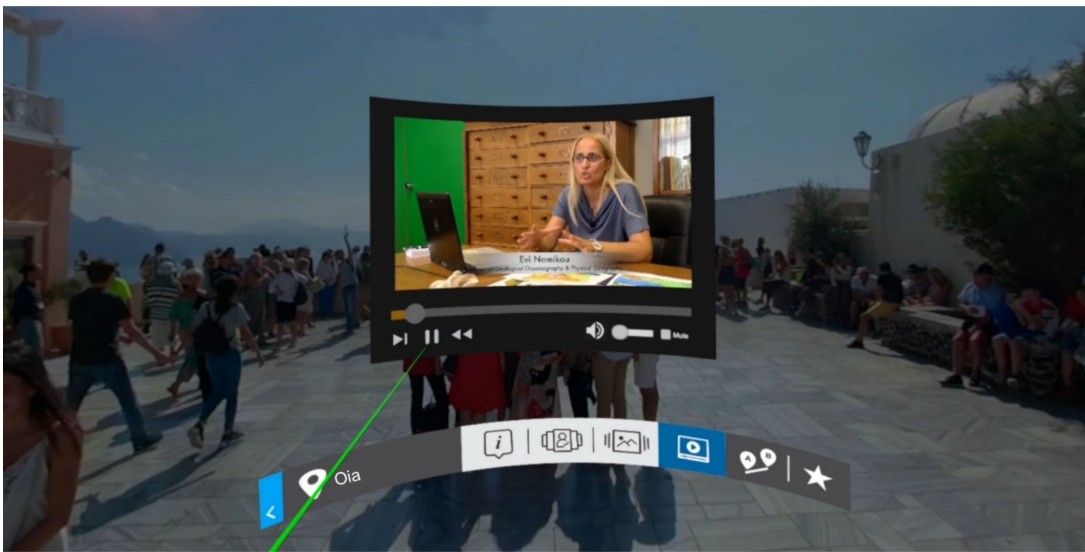

**Figure 16.** Walk Mode: Central square of Oia city—the Video Interaction Component is active.

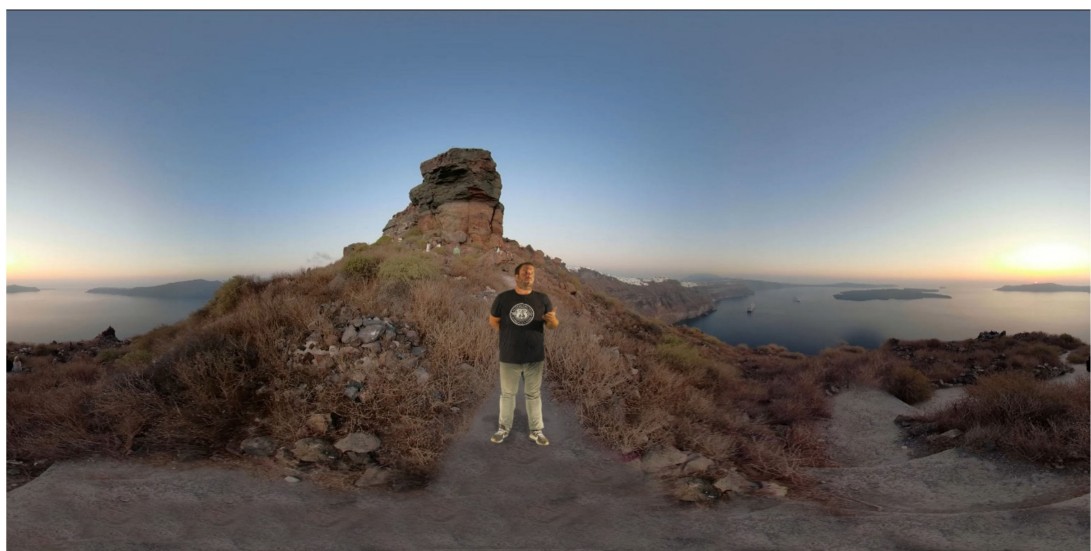

**Figure 17.** Guided tour by an expert—Green Screen 360° video production with a narrator (credits to Alexandros Arapantonis).

An additional choice is provided, on the interaction bar, for specially planned routes suggested by the VIRTUALDiver platform. This function provides a combination of POIs, based on their content, as well as a scheduled route for the user to visit (Figure 18). The user can easily navigate through those POIs and their Interaction Components without returning to the Fly Mode environment. The UI is designed in order to inform the user about the POI he/she is in at any specific time and provides indications of visited and not visited sites. By entering this mode, the interaction bar changes into a tour mode bar for easier navigation between the POIs.

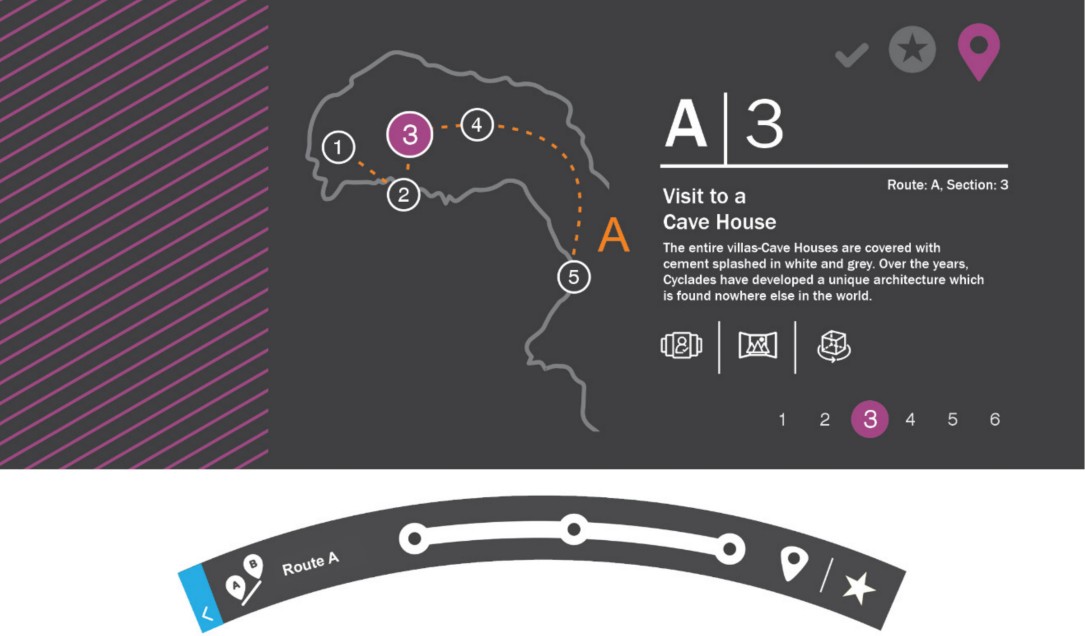

**Figure 18.** Specially planned route with the interaction bar changed into a tour mode bar.

The user moves to the next section, in a three-dimensional environment, specifically to the city of Oia (Figure 19). There he/she has the opportunity to move around, process the environment, and be informed about the geological peculiarity of the area. The photorealistic model of the city of Oia is combined with a sea simulation system, a high-resolution Skybox, and ambient sounds.

Navigation is achieved with VRTK's teleportation system, thus expanding the area in which the user can move without being limited by the physical space in which he/she is located. This also improves the user's sense of immersion, thus reducing the negative effects that can occur when driving in virtual environments, such as dizziness and nausea [54].

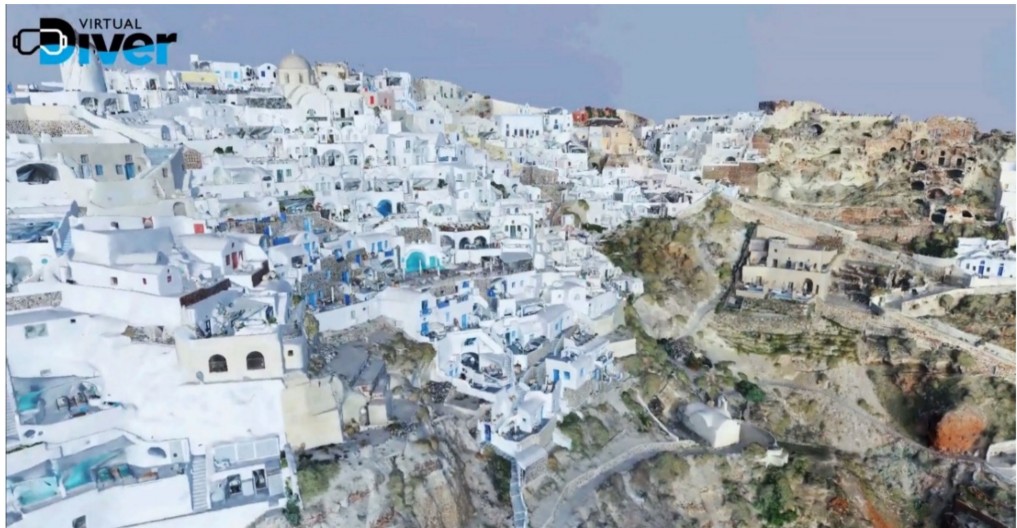

**Figure 19.** 3D environment of City of Oia. (3D Model has been produced by Up2metric VIRTUALDriver partners)

In order to move to the next section, the user interacts with a specially designed symbolic element and is transported in Diver Mode to an underwater environment (Figure 20). Here the user has the ability to navigate underwater, be informed about the flora and fauna, as well as interact with objects scattered on the bottom of the sea. Navigation in Diver Mode is similar to that in Fly Mode. The difference is that the speed is significantly reduced, compared to Fly Mode, to give a more realistic sense of underwater movement. In this environment, the user encounters fish and other marine species, which through artificial intelligence algorithms, move in space and around him/her. In addition, the user is able to choose which object he/she wants to interact with and gather more information about it, as well as hold and process specific three-dimensional objects, a function that increases realism and immersion.

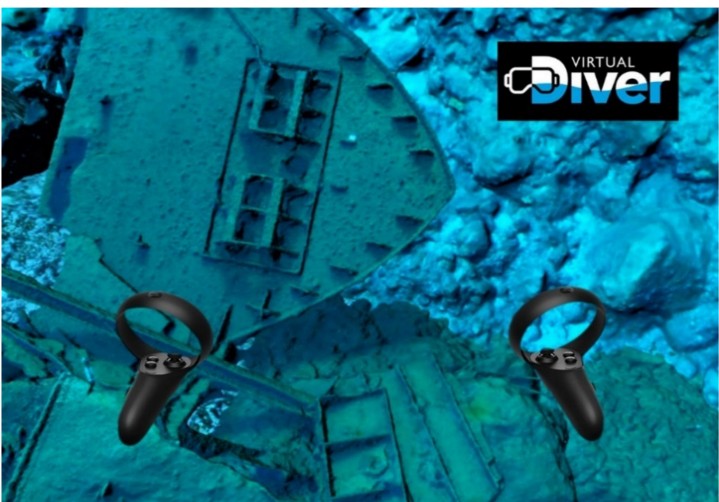

**Figure 20.** Diver Mode: User interacting with a 3D model of a shipwreck. (3D Model has been produced by Up2metric VIRTUALDriver partners)

Three-dimensional models, representing the sea bottom of Santorini, were used to set up the underwater scene realistically. These models come either from extensive research and development by the partners of the VIRTUALDiver consortium or have been acquired by other sources, e.g., digital libraries, to enrich the user environment. The main categories of models are:

- Natural objects such as stones, sand, tree trunks, shells
- Objects of the external environment
- Flora and fauna such as various species of fish, sea turtles, sea urchins, jellyfish, various species of algae, coral
- Objects of particular interest such as shipwrecks, rocks of geological value, and objects of narrative interest.

In addition to the 3D objects introduced on stage, lighting and sound effects were adjusted to enhance the realism of the underwater environment.

### 3.3. Discussion

The use of new research knowledge and innovative technologies to promote the Greek seabed, as well as free access to scientific data and the transfer of scientific knowledge to the general public, is now possible and can lead to the development of new touristic products, services, and activities, which can later attract tourists of general and/or special interest. At the same time, the introduction of Virtual and Augmented Reality technologies into a particularly interesting and hardly accessible underwater environment is a challenge for the niche market and creates new investment opportunities.

## 4. Conclusions

The main issue during the design process by a multidisciplinary team was the adoption of a commonly accepted way of managing multimedia material by all members, regardless of familiarity with programming or usage of the Unity 3D environment. By incorporating the use of visual scripting in the production process, VIRTUALDiver's methodology for implementing interactive experiences acquired an interdisciplinary character, enabling collaboration from the whole team. Although Unity 3D provides appropriate interface design tools, the production process required a corresponding specialization in the creation phases, on the one hand, and a significant implementation period on the other. In the VIRTUALDiver platform, the design of UI elements, as a production process, is taking place mainly in familiar design software (e.g., Adobe Photoshop [55]), and through special plugin modules, the produced material is introduced in the platform already marked with the respective interactivity properties. Following this methodology, the design time of the interfaces is reduced and the front-end management by the appropriate team (UX, UI designers, and content editors) is easily accessible.

Based on the adoption of the above the following design principles were followed:

- Easy navigation in the virtual environment.
- Easy access to content.
- Ability to navigate using digital maps (2D and 3D).
- Ability to navigate using virtual routes.
- Attractive interfaces of interactive experiences.

As a general statement, it can be said that although VR and AR account for dynamically developing fields, the necessary tools for virtual storytelling are still in an infant and experimental stage. Thus, the purpose of VIRTUALDiver is to make use of the current technological developments in VR and AR, to highlight the terrestrial and underwater wealth of Greece, and develop new types of cultural-touristic applications.

**Supplementary Materials:** The following are available online at http://www.mdpi.com/2076-3417/10/22/8172/s1, Video S1: VIRTUALDiver Project.

**Author Contributions:** Conceptualization, P.N., G.I. and G.P.; methodology, G.P. and K.M.; software, G.P. and K.M.; validation, G.P. and P.N., formal analysis, E.K and K.B.; investigation, G.P, K.M., A.T. and E.K..; resources, A.T. and K.B.; data curation, G.P.; writing—original draft preparation, G.P., K.B., A.T., and E.K; writing—review and editing, P.N.; visualization, G.I.; supervision, P.N.; project administration, V.A.; funding acquisition, P.N. All authors have read and agreed to the published version of the manuscript.

**Funding:** This research was funded by RESEARCH-CREATE-INNOVATE of the Competitiveness, Entrepreneurship and Innovation (EPANEK), NSRF 2014-2020, R.G. 15336 (Research project: VIRTUALDiver), grant number 02210.

**Acknowledgments:** We acknowledge funding from the "RESEARCH-CREATE-INNOVATE" of the "Competitiveness, Entrepreneurship and Innovation (EPANEK)", NSRF 2014-2020, R.G. 15336 (Research project: VIRTUALDiver) for the Department of Geology and Geoenvironment. We acknowledge Sarantinos Michalis, Maria Douza and Alexandros Arapantonis (STEFICON) and Christos Stentoumis, Ilias Kalisperakis (Up2metric) for their fruitful collaboration in the framework of the VIRTUALDiver project (www.virtualdiver.gr).

**Conflicts of Interest:** The authors declare no conflict of interest.

## Abbreviations

| | |
|---|---|
| AR | Augmented Reality |
| VR | Virtual Reality |
| UCH | Underwater Cultural Heritage |
| POI | Point of Interest |
| VRTK | Virtual Reality Tool Kit |
| UI | User Interface |
| UX | User Experience |
| GUI | Graphical User Interface |
| ICT | Information and Communication Technologies |
| NVMe | Non Volatile Memory Express |
| SSD | Solid State Drive |

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
