# Peer review of "The VIRTUALDiver Project. Making Greece’s Underwater Cultural Heritage Accessible to the Public"

_applsci, doi:10.3390/app10228172_

Round 1
Reviewer 1 Report
Well written and soundly explained. I would recommend to use more figures that illustrate underwater environment if possible.
Author Response
Dear Reviewer,
Thank you very much for your edits and your thorough review of our manuscript. We have made the spell check required and we have added Figure 11.
On behalf of all co-authors
Paraskevi Nomikou
Reviewer 2 Report
Dear Authors,
Thank for your paper entitled "The VIRTUALDiver Project. Making Greece’s 2 underwater cultural heritage accessible to the public" that I
found very interesting.
The paper is well written; however, few minor corrections are needed to enhance the readability of the text; these are listed below
(1) pl. add a list of all acronyms you used in the text;
(2) allow me to suggest the possibility to add a short video as electronic supplement to allow better evaluation of the outcomes of this research;
(3) figure caption is very short; you may add further details;
(4) I suggest adding some text labels on the figures;
(5) it is difficult to read the text the figures (for instance Fig. 1), background color is dark which reduces the clarity of the text
(6) Lines: 184-194:
-change First section to section 1 as written in figure 2;
(7) Lines: 254-257 are repeated on Lines: 260-262;
(8) Line 345: Scheme I change to Figure and include its reference in the text;
(9) pl. add a site map showing the Santorini Island. Although it is known it is better to add a map showing its location;
see for example Figure 10
(10) it is not mentioned anywhere in the text system requirements to visualize the site?
(12) Interested scientists, people can add comments and conserve them on their on PC?
to be clear did authors foreseen this possibility which is similar to what Google Earth do?
(13) line 408:
add Island to the caption and identify the meaning of the white solid circle;
(14) Lines 409, 418:
add text labels to the figures;
(15) Line: 429:
- better to change background color;
- add text to caption: 1).., 2)... etc
(16) Language compatibility of the site?
Reviewer
Author Response
Dear Reviewer,
Thank you very much for your edits and your thorough review of our manuscript. We have accepted nearly all your recommendations and we have made some additional improvements. Furthermore, below we list our feedback to your notes on the reviewed manuscript:
(1) add a list of all acronyms you used in the text;
According to the instructions for authors, abbreviations should be defined in parentheses the first time they appear in the abstract, main text, and in figure or table captions and used consistently thereafter. Nevertheless, we have added a list of acronyms just before the References section. We would like your response on that, as to whether you agree with this or you would like to us to place them somewhere else in the text.
(2) allow me to suggest the possibility to add a short video as electronic supplement to allow better evaluation of the outcomes of this research;
We have added additional material, and more specifically a supplementary video showcasing the functionality of the platform within the Unity environment.
(3) figure caption is very short; you may add further details;
We have added further details.
(4) I suggest adding some text labels on the figures;
Some figures are screenshots of the application and they show the design of the interface. With this in mind we cannot add labels or text on the figure.
(5) it is difficult to read the text the figures (for instance Fig. 1), background color is dark which reduces the clarity of the text
As some of the figures are screenshots, we cannot change the default colors of the system.
(6) Lines: 184-194:
-change First section to section 1 as written in figure 2;
We have made the changes according to your recommendation.
(7) Lines: 254-257 are repeated on Lines: 260-262;
We have deleted the repetition of the same sentence in section 3.1.1.
(8) Line 345: Scheme I change to Figure and include its reference in the text;
We have changed the word “Scheme” to “Figure” and included its reference in the text.
(9) pl. add a site map showing the Santorini Island. Although it is known it is better to add a map showing its location;
see for example Figure 10
We have added a map showing where the volcanic complex of Santorini is. In addition, in the inset map the area of interested is marked with an orange, rectangular outline.
(10) it is not mentioned anywhere in the text system requirements to visualize the site?
We have added the minimum requirements needed for the platform.
(12) Interested scientists, people can add comments and conserve them on their on PC?
to be clear did authors foreseen this possibility which is similar to what Google Earth do?
At this stage of the project, comments facilitate the workflow of the design team producing this experience for the application. For the time being, comments do not consist a tool for the end user. We will reexamine the possibility as we finalize the design of the product.
(13) line 408:
add Island to the caption and identify the meaning of the white solid circle;
This figure represents the interface of the application. Data like geographical names will be given to the user in other ways. We have explained what each graphic represents in the caption of the figure.
(14) Lines 409, 418:
add text labels to the figures;
Some figures are screenshots of the application and they show the design of the interface. With this in mind we cannot add labels or text on the figure.
(15) Line: 429:
- better to change background color;
- add text to caption: 1).., 2)... etc
Some figures are screenshots of the application and they show the design of the interface. With this in mind we cannot add labels or text on the figure. This is also a screenshot from the application interface design, colors, texts and labels are chosen to meet the application’s functional needs, user understanding and overall esthetic.
(16) Language compatibility of the site?
We have added this information in the text.
Additionally, we have made the spell check required and we have made improvements at the results.
Thank you again for reviewing our manuscript.
On behalf of the co-authors
Nomikou Paraskevi
Reviewer 3 Report
English language in the text of manuscript could be improved.
Author Response
Dear Reviewer,
Thank you very much for your edits and your thorough review of our manuscript.
We have made a thorough language editing.
We have incorporated additional bibliographic references and we have made improvements at the results.
Thank you again for reviewing our manuscript.
On behalf of the co-authors
Paraskevi Nomikou